# Luminescence- and Fluorescence-Based Complementation Assays to Screen for GPCR Oligomerization: Current State of the Art

**DOI:** 10.3390/ijms20122958

**Published:** 2019-06-17

**Authors:** Elise Wouters, Lakshmi Vasudevan, René A. J. Crans, Deepak K. Saini, Christophe P. Stove

**Affiliations:** 1Laboratory of Toxicology, Department of Bioanalysis, Faculty of Pharmaceutical Sciences, Ghent University, Ottergemsesteenweg 460, 9000 Ghent, Belgium; elise.wouters@ugent.be (E.W.); lakshmi.vasudevan@ugent.be (L.V.); rene.crans@ugent.be (R.A.J.C.); 2Department of Molecular Reproduction, Development and Genetics, Indian Institute of Science, Bangalore 560012, India; deepaksaini@iisc.ac.in

**Keywords:** G protein-coupled receptor (GPCR), dimerization, oligomerization, protein complementation assay, bimolecular fluorescence complementation (BiFC) assay, bimolecular luminescence complementation (BiLC) assay

## Abstract

G protein-coupled receptors (GPCRs) have the propensity to form homo- and heterodimers. Dysfunction of these dimers has been associated with multiple diseases, e.g., pre-eclampsia, schizophrenia, and depression, among others. Over the past two decades, considerable efforts have been made towards the development of screening assays for studying these GPCR dimer complexes in living cells. As a first step, a robust in vitro assay in an overexpression system is essential to identify and characterize specific GPCR–GPCR interactions, followed by methodologies to demonstrate association at endogenous levels and eventually in vivo. This review focuses on protein complementation assays (PCAs) which have been utilized to study GPCR oligomerization. These approaches are typically fluorescence- and luminescence-based, making identification and localization of protein–protein interactions feasible. The GPCRs of interest are fused to complementary fluorescent or luminescent fragments that, upon GPCR di- or oligomerization, may reconstitute to a functional reporter, of which the activity can be measured. Various protein complementation assays have the disadvantage that the interaction between the reconstituted split fragments is irreversible, which can lead to false positive read-outs. Reversible systems offer several advantages, as they do not only allow to follow the kinetics of GPCR–GPCR interactions, but also allow evaluation of receptor complex modulation by ligands (either agonists or antagonists). Protein complementation assays may be used for high throughput screenings as well, which is highly relevant given the growing interest and effort to identify small molecule drugs that could potentially target disease-relevant dimers. In addition to providing an overview on how PCAs have allowed to gain better insights into GPCR–GPCR interactions, this review also aims at providing practical guidance on how to perform PCA-based assays.

## 1. Introduction

Membrane receptors are the key players in mediating communication between the cell and the extracellular space. The G protein-coupled receptor (GPCR) protein family represents one of the largest group of cell membrane signaling proteins in the human genome [1]. Currently, approximately 800 members of this superfamily are classified according to the International Union of Pharmacology, Committee on Receptor Nomenclature and Classification (NC-IUPHAR). Herein, GPCRs are grouped into three classes, namely Class A (rhodopsin-like), Class B (secretin receptor family), and Class C (metabotropic glutamate-like) [2]. The remaining GPCRs—fungal mating pheromone receptors, cyclic AMP receptors, and frizzled/smoothened—are also included in alternative classification systems [3].

G protein-coupled receptors are activated by a panel of different triggers, including photons, nucleotides, ions, hormones, peptides, chemokines, and others. Notwithstanding the broad collection of ligands to activate specific GPCRs, they all share a common structure, namely, an N-terminal domain, seven transmembrane α-helices, extra- (ECL1–3) and intracellular loops (ICL1–3) and a C-terminal domain. Specific domains within these GPCRs show plasticity, since conformational changes are clearly distinguishable between their activated and inactive states, as observed from the crystal structures [4]. Upon activation, the motile domains, such as the ICL3 and the C-terminus, are crucial for interaction with intracellular signaling partners, i.e., G-proteins, GPCR kinases (GRKs) and β-arrestins. Given their crucial role in cellular signaling, altered activity or deviating expression levels of GPCRs have frequently been correlated with diseases, including neurodegenerative disorders, depression, and cancer [5,6,7,8,9,10].

Overall, their abundance, regulation in pathophysiology of diverse disease areas, accessibility at the cell surface, and availability of druggable binding sites have made GPCRs the largest drug target family. It is, hence, not surprising that 34% of the current FDA-approved drugs on the market target GPCRs [11]. Interestingly, virtually all pharmaceutical companies have primarily focused on GPCR monomers as a target for drug development. However, the recent knowledge that GPCRs form homo-oligomeric or hetero-oligomeric complexes has opened up unexplored avenues in the development of new drug therapies (e.g., small ligands, peptides, and bivalent ligands) [12,13,14,15].

### 1.1. GPCR Dimerization

There are approximately 150,000 to 300,000 protein–protein interactions in the human interactome [16,17] and constant efforts are ever-expanding this number. This also holds true for the interest in GPCR–GPCR interactions, as evidenced by the remarkable rise in publications in this field of research, with over 300 reports (Figure 1) covering GPCR multimers during the last 5 years.

While the traditional view on GPCRs was a “one ligand-one monomeric GPCR unit-one signaling protein” model, an emerging body of evidence shows that GPCRs can form homo- or hetero-oligomeric complexes. This phenomenon adds an additional layer to the complexity of GPCR signaling. Dysfunction of these dimers has been associated with multiple diseases, e.g., pre-eclampsia (angiotensin II type 1 and bradykinin receptor B2 heterodimer complexes) [18], schizophrenia (dopamine D2 receptor (D_2_R) heterocomplexes) [19], depression (serotonin 1A receptor 5-HT1A and serotonin 7 receptor 5-HT7 heterocomplexes, galanin receptor type 1 (GalR1), type 2 (GalR2), and 5-HT1A heterocomplexes) [20,21,22,23,24,25,26].

Although the concept of GPCR dimerization has been controversial over the past two decades, several sophisticated studies have led scientists to the validation of the GPCR dimerization theory. An indisputable and widely accepted example is the obligate dimer of Class C GABA receptors [27]. These receptors are obligatory dimers, as GABAB1 is necessary for ligand binding and GABAB2 ensures efficient cell trafficking and downstream signaling. Both protomers are not functional when expressed alone. In addition to obligate GPCR dimers observed in Class C receptors, an increasing amount of convincing evidence suggests there is also -transient dimer formation between Class A GPCRs [28,29,30]. Conflicting observations both in favor of and against oligomerization have been reviewed by Bouvier and Herbert (2014) [31] and by Lambert and Javitch (2014) [32]. Recent insights into Class A GPCR dimerization have shed new light on this concept, i.e., a large-scale comparative study of 60 members of the Rhodopsin family revealed that only a small fraction, i.e., 20%, forms dimers [33]. The dimer formation of Class A GPCRs seems to have a clustered distribution, and furthermore, a striking correlation has been suggested between receptor organization and size of the GPCR family. A restricted family size, as for the small Glutamate family, correlates with predominantly dimeric behavior. This also seems to be true for the receptors exhibiting less diversity like the frizzled GPCRs [33].

Besides the fact that Class A GPCRs seem to dimerize to a lesser extent as compared to Class C receptors, the dimerization event itself is also much more complicated, i.e., Class A GPCRs often seem to interact at multiple interfaces, thus causing a broad impact on functionality [34]. This is noted in the case for the muscarinic acetylcholine M_1_ or M_2_ dimerization, wherein reports have suggested that the sites of contact are transient and could involve multiple regions of the receptor [35,36,37]. For the D_2_R, this transient dimerization phenomenon has been studied in more detail, with a lifetime of 68 ms being assigned to the dimer. This dimer lifetime could be prolonged with a factor of approximately 1.5 by agonist addition (i.e., dopamine and quinpirole) [38,39], whereas incubation with the antagonist spiperone reduced the level of D_2_R homodimerization by more than 40% [40], indicating a potential role of the dimer in signaling cascades. Dijkman and colleagues introduced the “rolling dimer” interface model for the neurotensin receptor 1 homodimer. This concept unites earlier seemingly conflicting data since multiple dimer conformations could co-exist that interconvert during the dimer lifetime, by rotation of the monomers relative to one another.

In general, the phenomenon of oligomerization can exert a significant impact on receptor–ligand binding, downstream signaling, crosstalk, internalization, and trafficking. For instance, the interaction of D_2_R with dopamine D3 receptor (D_3_R) results in a higher potency of certain anti-parkinsonian agents, like pramipexole, compared to their monomers [28]. Furthermore, selective stimulation of D_1_R or D_2_R or both by the neurotransmitter dopamine triggered co-internalization of the D_1_R–D_2_R heterodimer [29].

### 1.2. Studying GPCR–GPCR Interactions: Biochemical Methods

In order to target and study GPCR–GPCR interactions, multiple in vitro (and in vivo) biochemical and physical, and functional studies have been reported in the literature, which described approaches such as co-immunoprecipitation (co-IP), proximity ligation assays (PLA), and fluorescence- or bioluminescence-based techniques [41].

The presence of multiple protein–protein interactions (PPIs) has been demonstrated by co-IP or pull-down assays [42,43,44]. On a non-reducing SDS-PAGE (sodium dodecyl sulfate-polyacrylamide gel electrophoresis), it has been reported that GPCRs often migrate in a way that corresponds to twice their molecular weight. Co-IP of endogenous GPCRs entirely relies on the selectivity of the antibodies used. Furthermore, this technique necessitates the disruption of biological samples and the solubilization of membrane proteins and generally provides no information about the physical interaction.

On the other hand, fluorescence- or bioluminescence-based energy transfer techniques can be applied in living cells, thereby overcoming some of the limitations of the classical biophysical techniques. For GPCR dimerization, this has been reviewed by Ciruela et al. (2010) [45]. Fluorescence and bioluminescence resonance energy transfer (FRET and BRET) involve the non-radiative energy transfer between a donor and acceptor, which allows the examination of PPIs in their native cell context by monitoring emissions from the acceptor species. Fluorescence resonance energy transfer microscopy offers the advantage that it allows visualization of the interaction inside living cells. Bioluminescence resonance energy transfer avoids some of the troubleshooting involved with FRET such as photobleaching, spectral bleed-through, etc., but does not provide spatial information on the interaction. Below, we will focus primarily on recent advances in yet another biochemical method, namely, bioluminescence and fluorescence-based complementation techniques that have been applied to study GPCR–GPCR interactions.

### 1.3. Protein Complementation Assays

Protein complementation assays (PCAs), also referred to as split systems, have been implemented over the past 2 decades to study PPIs. In these assays, a reporter protein with enzymatic or fluorescent properties is engineered or “split” into non-active or non-fluorescent fragments. These moieties are fused to potential interacting proteins. Upon interaction, the fragments will be brought into close proximity and re-assemble spontaneously into a functional biosensor. Although these assays do not formally prove direct protein–protein interactions, they do suggest co-localization of the GPCRs of interest since PCAs rely on the fusion partners “interacting”. Complementation-based assays comprise bimolecular fluorescence and luminescence complementation (BiFC/BiLC) as well as β-Lactamase and β-Galactosidase complementation assays. Since 1997, several “split” fluorescent and luminescent proteins have been developed, of which the most commonly used ones are shown in the timeline in Figure 2.

These assays have the potential to be implemented as screening assays to identify GPCR–GPCR interactions, which later on should be confirmed by a variety of complementary approaches, for example, Spatial Intensity Distribution Analysis (SpIDA), advanced microscopy techniques, and molecular dynamics, among others [35,39,46,47,48,49]. A benefit of a robust complementation assay is not only its operational simplicity but even more the possibility to test if dimers occur constitutively or whether ligands can alter the oligomeric state of GPCRs. Moreover, these cell-based assays lend themselves for early-stage drug discovery, since molecules that potentially exert an impact on the level of dimerization, whether desired or undesired, can be rapidly identified. In addition, off-target effects can be revealed, such as altered receptor expression, localization or cellular signaling. An overview of the available PCAs as of today and their characteristics are presented in Table 1.

Many questions remain unanswered in the field of GPCR di- and oligomerization. Therefore, it is a highly warranted first step to fully characterize the in vitro assays featuring robust sensitivity in detecting PPIs of interest. This review aims at offering insight into the progress that has been made in the field of protein complementation assays to study GPCR dimerization. This progress concerns, for example, optimization of complementation assays in stability and light output by a consensus sequence driven strategy. Several mutations applied in fluorescent and luminescent complementation assays in the field of GPCR dimerization will be discussed throughout the following sections of this review, for additional information concerning mutational optimization of split reporter genes in general, we refer to the excellent review by Wehr and Rossner [50].

## 2. Fluorescence-Based Complementation Assays

One of the well-studied PCAs is BiFC, which is based on the structural and functional complementation of two non-fluorescent protein fragments brought into close proximity by their interacting fusion partners (Figure 3). Upon complementation of the fluorescent protein, the fluorescence can be read via a plate-reader, imaged using a fluorescence microscope or even analyzed by flow cytometry without the need for any treatment of the cells.

In 1999, Regan and colleagues reported BiFC for the first time in *Escherichia coli* (*E. coli*) by using a strategy based on the non-covalent association of the split fragments of green fluorescent protein (GFP), fused to two antiparallel leucine zippers [51]. Subsequently, Hu et al. [52] reported on the development of a BiFC assay based on the split fragments of yellow fluorescent protein (YFP), tagged to two interacting transcription factors in living mammalian cells. Shyu et al. [53] developed Venus, one of the brightest fluorescent proteins, for studying PPIs at physiological conditions using BiFC. Since then, numerous types of fluorescence complementation-based assays have been developed to visualize signaling events involving two or even more interacting proteins, ranging from bacteria to mammalian cells.

### 2.1. Fluorescent Proteins

#### 2.1.1. Green Fluorescent Protein (GFP)

Regan and colleagues used GFP, the first fluorescent protein, for characterizing BiFC by tagging the split fragments of GFP, i.e., NGFP and CGFP, split at residue 157–158, to strongly interacting antiparallel leucine zippers. The split fragments did not self-assemble in bacteria, whereas upon fusion to antiparallel leucine zippers, the fragments could properly fold to form a functional GFP protein, and hence, gain fluorescence. A recent study performed by Son and colleagues [54] made use of enhanced GFP (EGFP), an improved version of GFP that is brighter and more photostable. The EGFP molecule was split to yield N-EGFP (1–158) and C-EGFP (159–238), respectively. These split parts were fused to α-factor receptor (Ste2p), to study homodimerization in *S. cerevisiae*. Both full length and C-terminally truncated forms of the receptors led to fluorescence, indicating dimerization.

A recent advance in BiFC based on GFP is Tripartite GFP, introduced by Cabantous et al. [55] in 2013 in an attempt to cope with the problem of poor folding and self-assembly seen with split GFP. Tripartite GFP is based on the tripartite association of two twenty amino-acid residue fragments of GFP, coined as GFP 10 and GFP 11, that can be fused to the proteins under investigation, and the complementary GFP 1–9 detector. Upon interaction between the protein partners, GFP 10 and GFP 11 associate with GFP 1–9 to form the functional GFP molecule. This system was initially characterized in *E. coli* and in mammalian cells, where the rapamycin induced association of FRB (FKBP-rapamycin binding domain of mTOR) and FKBP12 (FK506- and rapamycin-binding protein) was studied. This system has extensively been exploited for the validation of multiple reporter systems. Although use of this Tripartite GFP system has not been reported for studying GPCR dimerization, it could be an advantageous platform to be used over the conventional split GFP.

#### 2.1.2. Yellow Fluorescent Protein (YFP) Variants

In 2001, Nagai et al. [56] first demonstrated the use of the yellow fluorescent protein (YFP) in BiFC, wherein they used a variant of YFP to study the effect of Ca^2+^ on PPI in living cells. Shortly after, Griesbeck and colleagues [52,57] tried several permutations and combinations of split regions in EYFP (S65G, S72A, T203Y) to improve the fluorescence signal along with minimal self-assembly of the split fragments. Several variants of YFP have been developed such as Citrine (a variant having a Q69M mutation) and Venus (a variant carrying a F46L mutation), that folds even at a physiological temperature of 37 °C [58]. This led to the usage of these fluorescent protein fragments to study GPCR–GPCR interactions.

Vidi et al. (2008) [59,60] used split fragments of Venus fused to the adenosine A_2A_ receptor (A_2A_) to address the question about A_2A_ oligomerization in the CAD neuronal cell line. Co-expression of A_2A_-VN and A_2A_-VC in these cells, as well as in HEK293 cells, resulted in fluorescence, indicating A_2A_ dimerization, in contrast to negative controls wherein cells transfected with A_2A_-VN/D_1_R-VC/M_4_R-VC did not show any fluorescence, pointing at the specificity of the interaction.

Similarly, Kilpatrick et al. (2014) [61] assessed dimerization of a receptor for NPY, Neuropeptide Y Y1/Y5. This neuropeptide is a widely expressed modulator of the central nervous system, mainly known for its role in the regulation of appetite upon its release from hypothalamic arcuate neurons. The receptors, tagged with split YFP, showed heterodimerization and the irreversible nature of these fragment tags also helped to elucidate the cellular localization of these dimers. Evaluation of the effect of ligands on the internalization of dimers, as opposed to individual protomers, demonstrated that Y1/Y5 receptor dimers displayed altered ligand pharmacology, indicative of allosteric interaction.

Przybyla and Watts [62] in 2010 used Venus-based BiFC to demonstrate heterodimerization between the cannabinoid receptor 1 (CB_1_) and the dopamine D_2Long_ receptor (D_2L_R) in CAD cells, wherein CB_1_ and D_2L_R receptors were fused at their C-terminus to VN and the C-terminal fragment of Cerulean (CC), respectively (see Section 2.2.1).

Ang et al. [63] in 2018 assessed homo- and heterodimerization of free fatty acid receptors, GPCRs that are expressed on mammalian cells to sense the short-chain fatty acids derived from the microbiota. To study dimerization, HEK293T cells were transfected with VN- or VC-tagged (free fatty acid receptor 2/3) FFAR2/3 receptor. Homodimerization of FFAR3 and FFAR2, as well as heterodimerization between FFAR2-VN and FFAR3-VC were observed, as compared to negative control FFAR2/3 tagged to VN or VC plus P2RY1-VN or P2RY1-VC (for heterodimerization) and P2RY1-VN plus P2RY1-VC (for homodimerization).

Xue et al. [64] in 2018 used a BiFC assay based on split Venus to detect heterodimerization between growth hormone secretagogue receptor 1α (GHSR1a) and Orexin 1 receptor (OX1R). Recently, Navarro et al. [65] (2018) used interfering peptides corresponding to regions in TM1–7 domain for A_2A_ and TM5–7 domain for A_1_, each fused to TAT peptides, to disrupt the interaction between the A_1_–A_2A_ heteromer. They therefore expressed receptors tagged with split fragments of YFP (nYFP and cYFP) in HEK293T cells. In the absence of the TM peptides, fluorescence complementation was detected for A_2A_-nYFP and A_2A_-cYFP in the presence of untagged A_1_, suggesting homodimerization of A_2A_. The interaction sites were mapped to TM4 and 5 of A_2A_ with the help of interfering peptides. In the same way, a fluorescence signal corresponding to the heterodimer (A_1_-nYFP and A_2A_-cYFP) was also detected. Upon addition of TM peptides corresponding to regions 5/6, a reduction in fluorescence was observed, suggesting the involvement of these regions in the formation of the heterodimer. In the same year (2018), BiFC using split fragments of YFP was utilized by Hinz et al. [66] for studying homodimerization between A_2A_ and heterodimerization with A_2B_ receptor. For this purpose, CHO cells were transfected with a constant amount of A_2A_-NYFP and increasing amounts of HA-A_2A_-CYFP and a strong fluorescence signal was observed, indicative of the A_2A_-A_2A_ homodimer. For the purpose of studying A_2B_-A_2A_ heterodimerization, the GPCRs were tagged with NYFP and CYFP, respectively. CHO cells transfected with constant amounts of A_2B_-NYFP and increasing amounts of HA-A_2A_-CYFP showed significantly higher fluorescence when compared to the negative control comprising of GABAB2-NYFP and increasing amounts of HA-A_2A_-CYFP.

An interesting study was performed by Song et al. [67] in 2019, where a homo-molecular fluorescence complementation (homo-FC) probe was designed by splitting a single fluorescent protein (superfolder GFP, sfGFP) into two fragments and linking the C-terminal fragment (strands 8–11) with a short linker N-terminally to the N-terminal fragment (strands 1–7). This “flopped fusion” construct, which does not show intramolecular self-complementation, was used to tag the β_2_ adrenergic receptor (β_2_-AR), to study its homodimerization. A fluorescent signal was generated by complementation of strands 1–7 of one “flopped” sfGFP molecule, fused to one β_2_-AR, with strands 8–11 of another “flopped” sGFP molecule, fused to a second β_2_-AR. The advantage of this approach lies in the fact that only one receptor construct needs to be generated, to allow the assessment of homodimerization.

#### 2.1.3. Cyan Fluorescent Protein (CFP)

Apart from using the aforementioned split Venus, Vidi et al. [59,68] also used split fragments of Cerulean to show that A_2A_ homodimerizes in CAD cells (see Section 2.2.1). The same fluorescent protein fragments were also used in 2010 by Przybyla and Watts [62] to illustrate the homodimerization of D_2L_R.

#### 2.1.4. Red, Far-Red and Near-Infrared Fluorescent Proteins

Red fluorescent proteins encompass a couple of fluorescent proteins that emit in the red region of the visible spectrum. They have evolved through time to overcome issues such as slow maturation or the tendency to oligomerize (as exhibited by Discosoma Red (DsRed)) [69]. Red variants such as monomer red fluorescent protein 1 (mRFP1) are not appropriate for BiFC due to their low extinction coefficient, quantum yield, and low photostability, despite faster maturation than DsRed and better tissue penetration. Although several variants of RFP have been used for FRET, BiFC assays with the split parts have not been reported for demonstrating GPCR–GPCR interactions.

For imaging deep tissues in animals, fluorescent proteins with emission in the far-red region of the spectrum are chosen. RFP, mcherry and DsRed, have limited application in BiFC due to the requirement of low temperatures for their maturation [70,71]. This led to the development of mKate, a far-red, monomeric form of Katushka fluorescent protein, that matures faster and is photostable [72]. BiFC based on split fragments of mKate fused to transcription factors was successfully applied in COS-7 cells [73], but has not been applied yet for studying GPCR oligomerization.

### 2.2. BiFC Assays

#### 2.2.1. MBiFC

Multicolor BiFC (MBiFC) has gained a lot of attention in recent years as it essentially gives the freedom to explore multiple protein–protein interactions inside the same living cell [74]. It has been studied with split parts of Venus and Cerulean, wherein the C terminal part of Cerulean is tagged to protein A (CC), the N-terminal part of Venus is tagged to protein B (VN), and finally, the N-terminal fragment of Cerulean is fused to protein C (CN). The C-terminal fragment of Cerulean (CC), comprising residues 155–238 (C155), can functionally complement with the N-terminal fragment of VN and CN, comprising residues 1–172 (N173), to produce a Venus (VN+CC) signal or Cerulean (CN+CC) signal. Since these parts are non-fluorescent per se, simple employment of different excitation and emission wavelengths allows to study simultaneously the interaction between A–B and A–C within the same cell [59,75,76].

Multicolor BiFC has aided in the understanding of GPCR oligomerization amongst all its other applications. Vidi et al. [59] (2008) used MBiFC to visualize the hetero- and homodimerization capacity of A_2A_ in the CAD neuronal cell line. To this end, D_2L_R-VN was co-transfected with A_2A_-CN and A_2A_-CC in CAD cells. The detection of Venus (D_2L_R-VN/A_2A_-CC) and Cerulean (A_2A_-CN/A_2A_-CC) fluorescence signals revealed the presence of the A_2A_–D_2L_R heteromer and A_2A_ homomer, respectively. In addition, fluorescence microscopy also revealed that the dimers were localized at the plasma membrane as well as in intracellular compartments.

Przybyla and Watts [62] in 2010 showed oligomerization of CB_1_ and D_2L_R. To achieve this, they used CAD cells that expressed CB_1_-VN and D_2L_R fused to the split fragments of Cerulean (D_2L_R-CN, D_2L_R-CC) and these cells were imaged using fluorescence microscopy. The heterodimers in the cell formed by CB_1_-VN–D_2L_R-CC produced a Venus signal that was mainly localized in the intracellular compartment, whereas the homodimers of D_2L_R produced a Cerulean signal that was localized both in the intracellular compartment and on the plasma membrane.

#### 2.2.2. BiFC-RET

The fluorescent proteins used in BiFC could also be coupled to FRET or BRET to study GPCR oligomerization, e.g., functionally complemented YFP could act as a FRET/BRET acceptor. BiFC-BRET [77] employs the energy transfer between GPCR-A fused to a luciferase and the complemented fragments of YFP that are fused to GPCR-B and GPCR-C that are as close as 10 nm apart, indicating a trivalent complex of GPCRs.

Navarro et al. [78] in 2008 used BiFC-BRET to demonstrate the interaction between D_2_R, A_2A_, and CB_1_ receptors in HEK cells. The split fragments of YFP fused to CB_1_ and A_2A_ acted as the acceptor fluorophore in BRET, and *Renilla* luciferase (*R*Luc) fused to D_2_R (D_2_R-*R*Luc) acting as the donor.

BiFC-FRET [79] is similar, offering the additional benefit that the complex can be visualized within the cell. BiFC-FRET was used to understand the existence of higher order oligomers in the case of A_2A_ by using fusion constructs with Cerulean and split parts of Venus. Prior to this, to test if the system could record FRET, A_2A_ dimerization was tested by fusing A_2A_ to Cerulean and Venus. Based on this, CAD neuronal cells were co-transfected with A_2A_-Cerulean and A_2A_ fused to split parts of Venus. FRET signals were detected, suggesting the existence of A_2A_ oligomers, as opposed to a negative control wherein cells transfected with M_4_ as the acceptor only showed very low FRET [60].

In 2017, Bagher et al. [80] combined BRET with BiFC which forms the basis of “sequential resonance energy transfer 2” (SRET^2^). Herein, the donor for BRET was the *Renilla* luciferase (*R*Luc) fused to D_2L_R (D_2L_R-*R*Luc), which upon addition of the substrate coelenterazine 400a, excited the acceptor GFP^2^ fused to D_2L_R (D_2L_R-GFP^2^), thereby confirming the homodimerization of D_2L_R. Sequentially, the GFP^2^ acted as FRET donor for the CB_1_-VN/CB_1_-VC homodimer, thus providing evidence for heterotetramerization. With increasing concentrations of CB_1_-VN/CB_1_-VC, there was a hyperbolic increase in net SRET^2^, as opposed to negative control cells that were transfected with D_2L_R-*R*Luc and mGLuR6-GFP^2^ and increasing concentrations of CB_1_-VN/CB_1_-VC, confirming that the homodimers of D_2L_R and CB_1_ associate to form oligomers.

### 2.3. Ligand-Dependent Modulation of Dimerization

The level of GPCR dimer formation could possibly be altered by ligands interacting with the receptor(s). The potential of split-protein sensors to monitor the dynamic changes in dimerization, provoked by chemical inducers or inhibitors, was demonstrated using the rapamycin-dependent FKBP/FRB Chemically Induced Dimerization (CID) system [81]. The first demonstration for the use of MBiFC to monitor drug-modulated GPCR oligomerization was published by Vidi et al. [59] in 2008 for the A_2A_–D_2_R interaction. Stimulation of D_2_R with the D_2_R agonist quinpirole led to internalization of D_2_R homodimers as well as of A_2A_ − D_2_R oligomers, which was blocked by the D_2_R antagonist, sulpiride [59]. Furthermore, treatment with quinpirole decreased the formation of A_2A_ − D_2_R heterodimers, as compared to A_2A_ homodimers. Also, treatment with 5-N-methylcarboxamidoadenosine (MECA), an agonist for the adenosine receptor, increased the proportion of A_2A_/D_2L_R heterodimers, as compared to homodimers of A_2A,_ while treatment with its antagonist (CGS15943) had no effect on BiFC. Interestingly, these drug-induced alterations on the formation of oligomers could not be supported by changes in the receptor density.

As discussed above, Przybyla and Watts [62] demonstrated heterodimerization of CB_1_ − D_2L_R using MBiFC. Also, the influence of ligands on the balance between hetero- and homodimers was analyzed. Quinpirole and CP55,940 shifted the balance towards the formation of CB_1_ − D_2L_R heterodimers, as opposed to D_2L_R − D_2L_R homodimers. The D_2_R antagonist sulpiride favored D_2L_R homodimerization. Consequently, it was shown that sustained treatment with quinpirole influenced the expression of D_2L_R, which in turn might affect the dimerization status inside the cell as: D_2L_R − CB_1_ > D_2L_R − D_2L_R > A_2A_ − D_2L_R. In a similar way, studies also indicated that a CB_1_ antagonist (1-(7-(2-chlorophenyl)-8-(4-chlorophenyl)-2-methylpyrazolo (1,5-a)-(1,3,5)triazin-4-yl)-3-ethylaminoazetidine-3-carboxylic acid amide benzenesulfonate) could have a potential therapeutic role in Parkinsonism by enhancing the activity of L-DOPA [82].

In 2018, Navarro et al. [65] evaluated if activation of A_1_ or A_2A_ alone or by a combination of agonists could modulate the TM interface of the heteromer formed by A_2A_-nYFP and A_1_-cYFP. Selective agonists for A_1_, N6-cyclopentyladenosine (CPA), and A_2A,_ 4-(2-((6-Amino-9-(N-ethyl-β-D-ribofuranuronamidosyl)-9H-purin-2-yl) amino) ethyl) benzenepropanoic acid (CGS-21680), were used alone or in combination. None of the agonists, either used in combination or alone, could modify the TM interface of the A_1_ − A_2A_ heterodimer.

Various ligands were tested for their capacity to affect β_2_AR oligomerization by Song et al. [67] in 2019. A panel of compounds consisting of five agonists, two antagonists, and two inverse agonists were screened using the abovementioned homo-FC assay. They observed that agonists induced oligomerization, while inverse agonists reduced the signals, and a combination of both also resulted in reduced intensity of signal, thus confirming the same.

## 3. Luminescence-Based Complementation Assays

Luciferases, which have seen an expansive growth in use as reporter proteins in biological research, are attractive due to the high signal-to-background ratio associated with their usage, as no excitation light is required to generate a signal [83]. Similar to fluorescent proteins, these enzymes can also be used in applications where the luminescent protein itself is split into two fragments, which are conjugated to proteins of interest (Figure 4). Several luminescent proteins have served this purpose, with Firefly luciferase (FLuc) [84,85] and *Renilla* luciferase (*R*luc) [86,87] being the two most commonly used bioluminescent proteins. However, many other novel luciferases have also been developed with favorable characteristics in terms of stability, substrate requirement, brightness, and emission spectrum, e.g., *Gaussia princeps* luciferase (*G*Luc) [88] and NanoLuciferase [89,90,91]. All mentioned split-protein reporters were initially validated using the aforementioned FKBP/FRB CID system [84,88,92,93,94].

### 3.1. Renilla/Firefly Luciferase

FLuc is the least optimal reporter for employment as a bioluminescent tag, due to its size (±60 kDa) and its dependence on ATP, molecular oxygen, and magnesium for activity. Luciferases that use coelenterazine as a substrate, such as the luciferase from the sea pansy *Renilla reniformis R*Luc [95,96], have an advantage over FLuc in that they are not ATP dependent and only require the presence of molecular oxygen for the enzyme-catalyzed conversion of a substrate to a luminescent reaction product. As the bioluminescent activity of formerly used non-truncated luciferases is a limiting factor to permit their application in high throughput screenings, specific mutations of certain luciferases, like in *R*Luc, were selected using a consensus sequence-driven strategy and screened for their ability to confer stability in activity as well as for their light output [97]. 

For GPCR dimerization purposes, the split-luciferase reporter approaches have not been applied to the same extent as their fluorescence counterparts. Nevertheless, these reporter systems also allow the rapid detection of macromolecular GPCR–GPCR interactions. For instance, the split FLuc methodology was used in cell culture models as well as in tumor xenograft models of breast cancer (see Section 8) to measure changes in chemokine receptor CXCR4 (and CXCR7) homodimerization in response to pharmacological agents [98]. Likewise, the CXCR4 homodimer has also been demonstrated with a split *R*Luc assay [99]. For this *Renilla* luciferase complementation assay, *R*lucII was implemented, which is derived from *R*Luc, wherein two mutations (C124A and M185V) were introduced to make it brighter.

For studying dopamine D_2_R oligomers, an alternative *R*Luc construct was implemented, namely, *R*luc8 [100]. Owing to eight favorable mutations, this *R*Luc8 has a 4-fold improved light output, compared with the parental enzyme [97]. More recently, the same complementation assay was also used by Casado–Anguera et al. [101] (2016) to demonstrate the existence of a therapeutically relevant GPCR dimer, namely, the A_2a_ − D_2_R dimer. This dimer and its inter-protomer allosteric mechanisms have been proposed as a new model that could contribute to our knowledge concerning drug dosage for the treatment of Parkinson’s disease.

### 3.2. NanoLuciferase

One of the most recent and undoubtedly the most optimized luminescent PCA so far is “NanoLuciferase Binary Technology” or the NanoBiT^®^ system, developed by Promega. The assay is based on complementation of the split fragments of NanoLuciferase or NanoLuc. This engineered luciferase reporter is a small (19 kDa), ATP-independent luminescent protein, originating from a luminous deep-sea shrimp, *Oplophorus gracilirostris* [102]. In combination with the development of a novel cell-permeable imidazopyrazinone substrate, furimazine, this bioluminescence system generates a glow-type luminescent signal that is over 150-fold greater compared to that of the former *Renilla* and Firefly luciferases. NanoLuc exhibits high physical stability, in a wide range of environmental conditions such as temperature, pH, urea, and ionic strength [103]. As opposed to luminescent protein fragments described earlier, the NanoBiT^®^ subunits do not consist of two fragments similar in size but correspond to a small 1.3 kDa subunit (Small BiT; SmBiT) and an 18 kDa subunit (Large BiT; LgBiT). Both subunits have been thoroughly characterized and have a low affinity (K_D_ = 190 µM) for one another, thus providing the ability to follow kinetics of PPIs.

Although the NanoBiT^®^ PCA has only recently been developed [92], its broad applicability has proven successful in numerous research fields. NanoBiT^®^ has, for instance, been applied to develop bio-assays based on the recruitment of β-arrestins [104,105,106,107,108,109,110] and G-proteins [111] for the detection or activity profiling of certain compounds or to elucidate the molecular interaction between the transducer and a GPCR. For GPCR dimerization purposes, NanoBiT^®^ has been implemented to detect ligand-dependent modulation of D_2L_R − D_2L_R homodimers [40,112]. The methodology has also been used to perform live-cell monitoring of the dynamics of the interaction between the Melanocortin 4 receptor (MC4R), and the Melanocortin 2 receptor accessory protein 2 (MRAP2), as well as of MC4R homodimerization [113].

The NanoBiT^®^ system has been further optimized into a tri-part protein fragment complementation assay by two independent groups [114,115]. For this purpose, the LgBiT was dissected into a smaller C-terminal part of 11 amino acids (LcBiT) and an N-terminal derivative (LnBiT), serving as a “detector” protein of 16.5 kDa, similar to the Tripartite GFP mentioned in Section 2.1.1. Initially, this assay was designed to facilitate the purification of fusion proteins to procure significant quantities and to avoid the lapse of detection of the PPI of interest due to the steric hindrance of LgBiT, due to its higher molecular weight. In addition, another novel split-luciferase reporter based on NanoBiT^®^ was developed that implements GFP- and mCherry-recognizing nanobodies fused to LgBiT and SmBiT [116]. Using this experimental set-up, GFP- or mCherry-tagged dimers or higher order oligomers can be detected. As fluorescently tagged proteins are already often in use, this luminescent PCA can be directly used and consequently bypasses the tedious protein re-cloning to explore the different possible configurations for setting up a PCA (also see Section 6). Furthermore, due to the strong affinity of the GFP- and mCherry-recognizing nanobodies, a low limit of detection of PPIs or protein aggregations, from micromolar up to low nanomolar, can be achieved.

### 3.3. BiLC-RET

The application of BiLC has already shown its efficiency in combination with a BRET assay, allowing to detect ternary protein complexes. Sahlholm et al. [117] (2018) demonstrated, by combining BiLC of D_2_R − A_2a_ heterodimers with a YFP-tagged β-arrestin, a BRET signal could be obtained. Furthermore, it was postulated that D_2_R agonists quinpirole or UNC9994 require the formation of D_2_R − A_2a_R heterodimers, to promote β-arrestin2 recruitment.

## 4. Combinatorial Assays: BiFC and BiLC

For the detection of higher order oligomerization of GPCRs, a combinatorial assay with both BiFC and BiLC can be implemented (Table 2). Rebois et al. [77] (2008) introduced the concept of detection of a tetravalent complex using a combination of BiFC and BiLC, wherein four β_2_ARs were fused to split parts of Venus or *G*Luc and the homodimerized β_2_AR-*G*Luc acted as the donor for the homodimerized β_2_AR-Venus acceptor, the presence of a BRET signal indicating the presence of the tetramer. In the same year (2008), Guo and colleagues [100] showed the existence of a D_2_R homo-oligomer, using the same technique.

A complemented donor–acceptor resonance energy transfer (CODA-RET) assay showed that the A_2A_ − D_2_R dimer not only forms dimers but can also assemble to form a heterotetramer, composed of two receptor homodimers [101,118]. For this experimental setup, a complemented YFP was implemented and combined with *R*luc8.

## 5. Comparison of Split Protein Approaches

Protein complementation techniques have shown their applicability in many fields to unravel PPIs. However, every technique has its advantages and limitations, as summarized in Table 3 for BiFC and BiLC.

### 5.1. Advantages of PCA

A key advantage of BiFC and BiLC lies in the fact that complementation assays offer a robust and straightforward approach to evaluate PPIs in living cells. Consequently, their applicability in high-throughput experiments is a valuable quality. The development of MBiFC, BiFC-RET, or CODA-RET has made the detection of multiple protein–protein interactions feasible, such as GPCR oligomers, without the need for cell lysis or fixation. Due to the simplicity of performing these assays, they can be used as a screening platform for drugs [59]. When used with a plate reader, BiFC and BiLC also offer quantitative and rapid results on relatively large cell populations.

The most valuable aspect of BiLC is that kinetic measurements can be performed due to its sensitivity and the limited propensity of self-association events, at least for certain BiLC reporters (e.g., NanoBiT^®^). For the purpose of studying GPCR dimerization, the NanoBiT^®^ assay has been compared to related bioluminescence and fluorescence PCAs, including split Venus and *R*Luc [112]. As a benchmark, the generally accepted D_2_R − D_2_R dimer was compared in the different PCAs. The NanoBiT^®^ assay presented the best signal-to-noise ratio and was considered the most optimal candidate assay for targeting GPCR dimers. In addition, this method can also be implemented to analyze the kinetics of ligand-dependent modulation of dimerization, broadening its application potential [40]. Even high-throughput screenings can be performed, which is highly relevant, given the growing interest and effort to identify small molecule drugs that can target disease-relevant dimers (or even selectively alter GPCR dimer function). Moreover, BiLC has also proven its applicability in an in vivo setup [85,87].

On the other hand, BiFC has the capacity to visualize and localize the interaction, also offering insight into potential protein aggregation artefacts inside the cell. BiFC does not require any exogenous stains or substrate. Thus, depending on the intrinsic fluorescence, it allows the direct measurement of the interaction between proteins.

### 5.2. Limitations of PCA

BiFC and BiLC also come with their own set of drawbacks. First of all, both techniques require the fusion of the target proteins with split fragments of a fluorescent or luminescent protein, which could affect the original proteins’ interaction dynamics. Moreover, the amount of proteins expressed by the cell may potentially cause false positive readouts. Therefore, a good set of negative controls is obligatory (see Section 6.3). In addition, one should bear in mind that these techniques, though indicating the close proximity of the proteins being studied, are not an actual assessment of the physical contact between these proteins.

BiFC needs a longer maturation time, has an irreversible nature of complementation, shows a high degree of self-assembly of the split fragments of the fluorescent protein, and requires molecular oxygen for the maturation of the fluorophore, thereby making it unsuitable for obligate anaerobes. Since maturation is a time-dependent process, the data procured from BiFC are not in real time. The irreversibility of the technique also makes it more difficult to study the dynamics of interactions inside the cell and the influence of drugs on the interaction. Despite these limitations, BiFC is still the only protein complementation technique that gives information about the location of PPIs in living cells. For BiLC, its limitations are the need of a substrate and the rather limited possibility to detect the localization of the interaction.

## 6. Guidelines to Perform Accurate PCA-Based Assays

A potential but inherent drawback of PCA-based assays is the requirement to fuse fragments derived from fluorescent or luminescent proteins to the GPCR, which inherently may alter the GPCR’s function or cellular localization. Therefore, a list of recommendations for proper implementation of PCA-based assays can be put forward: (1) examine all possible combinations of fusion proteins; (2) verify the functionality and localization of the fused proteins; (3) include proper controls, i.e., non-interacting partners, to control for self-assembly; (4) include a normalization reporter, to allow compensation for differences in transfection efficiencies; (5) preferentially transfect an amount of DNA near to endogenous expression levels of GPCRs, to avoid false positives due to the random collisions or potentially generate stable cell lines; and (6) follow the kinetics of the GPCR interaction (Figure 5).

### 6.1. Possible Fusions

The development of PCAs requires multiple rounds of cloning, as one needs to design the most optimal “split” system for each GPCR of interest. Several configurations should be explored, being (i) different reporter systems, (ii) (N- or) C-terminally tagged proteins, and (iii) diverse linkers (i.e., length and composition). While this is a time-consuming step, having a certain workload, it is of paramount importance for the development of a robust, sensitive, and successful complementation assay.

#### 6.1.1. Selection of the Reporter System

A broad variety of fluorescent and luminescent split reporter fragments is available to unravel PPIs. For studying GPCR dimerization, different split reporter systems might be advised, depending on the characteristics of the GPCR dimer of interest (if known). For obligate GPCR dimers, the rules could be less stringent for the choice of the reporter system to be used, whereas for transient GPCR dimers, a dynamic reporter assay with kinetic measurements close to real-time is required. When BiFC is preferred, the truncated YFP fragments YN155 and YC155 are recommended, due to the high signal-to-noise ratio [119]. Venus, a brighter version of GFP, split at 155 or 173, is also commonly used for studying protein interactions. The benefit of using Venus fragments over YFP is that the complementation can be read at 37 °C, thus avoiding the incubation at 30 °C which is necessary when dealing with split-YFP.

When BiLC is preferred, the NanoBiT^®^ system offers advantages in flexibility and in the low affinity of the SmBiT–LgBiT fragments for each other, so dynamic interactions can be monitored in living cells in real time [92].

#### 6.1.2. (N- or) C-Terminally Tagged GPCRs

The site at which the tag is fused to a GPCR plays an important role in the functionality of the GPCR under study and the purpose of the study (Figure 6). All the published research on PCAs with GPCR dimers involves fusion proteins where the split biosensors are tagged to the C-terminal end of GPCRs, so transport and expression at the membrane surface are not hampered. Especially for monitoring interactions of Class C GPCRs, like the metabotropic glutamate receptors, C-terminal tagging of biosensors is recommended, as modification of the long N-terminus of the receptor could lead to deviating outcomes.

On the other hand, split fragments tagged to the N-termini of GPCRs, though not yet implemented for research on GPCR dimerization, might be an interesting tool as well, as these fusion proteins may enable the simultaneous monitoring of GPCR-GPCR interactions and the signaling properties of the dimer. For example, when a split biosensor is fused to the N-terminus of the GPCRs to monitor the GPCR-GPCR interaction, another split biosensor fused to the C-terminus of the GPCR can be applied to monitor G-protein or β-arrestin recruitment.

#### 6.1.3. Linkers

GPCRs of interest and the fragments of the split luminescent or fluorescent partners are usually separated by a linker sequence to assure the flexibility required for the proper folding and “settling” of the biosensor. Multiple linkers with alternating sequences and conformations have been designed, the design of these linkers often being empirical but based on linker sequences derived from natural multi-domain proteins [120].

The most frequently applied linkers in PCAs are composed of non-polar glycine and polar serine amino acids (“GS” linker), developed by Argos [121]. The flexibility of these linkers originates from the incorporation of these small residues. Moreover, the use of serine (or threonine) residues in these linkers contributes to the stability in aqueous conditions by the formation of hydrogen bonds with water molecules [122]. A model linker design involves the (GGGGS)_n_ template, where n represents the copy number. By adjusting this copy number, one might obtain the most optimal “GS” linker length to achieve appropriate separation of the GPCR and the split biosensor. For PCA purposes, often utilized GS linkers comprise roughly eight to fifteen amino acids, such as DGGSGGGS [123], GGGSGGGS [99,124] or GSSGGGGSGGGGSSG [105]. Other -more rigid linkers have also been implemented in PCA, for example ATGLDLELKASNSAVDGTAGPVAT [117]. The proline residues in linkers might increase stiffness to keep fusion moieties at a distance, whereas lysine residues are often added to improve solubility [122]. Furthermore, proper codon usage (i.e., avoiding rare ones) is advised to not compromise expression levels. Also, depending on the expression system, sequences which could represent potential protease recognition sites should be avoided.

Overall, the selection of the linker length is crucial to allow proper folding and accomplish optimal biological activity of the fusion proteins. For fusion proteins, the linker length is an essential feature of the PCA setup since linkers that are too short often result in impaired biological activity due to the inability to accommodate the complementation of the two split proteins, whereas linkers that are too long can create false positives. Therefore, the linker length will vary on a case-by-case basis. With this design in mind, the NanoBiT^®^ system offers different restriction sites in the provided biosensor templates, resulting in various GS linkers, ranging from 15 to 21 amino acids [112]. Some examples of linkers that have been applied in PCA assays to detect GPCR dimers are given in Table 4.

### 6.2. Functionality and Localization of the Fusion Proteins

Occasionally, the fusion of biosensors to a protein of interest (POI) may interfere with the expression level, activity or function of the latter [126,127]. This may, for example, highly depend on the sequence and length of the linker between the POI and the biosensor (as discussed in Section 6.1). To verify whether the functional integrity of the fusion POI is not affected, several techniques should ideally be implemented, such as ligand binding as well as functional assays (e.g., calcium signaling, cAMP, β-arrestin recruitment, MAPK activation, etc.). An immunofluorescence assay can also be used to confirm correct localization of the recombinant proteins at the plasma membrane.

### 6.3. Non-Interacting Partners

Studying GPCR–GPCR interactions by complementation assays is relatively straightforward. However, the importance of including appropriate controls (i.e., GPCRs that do not bind to the GPCR of interest) should not be underestimated. It is also essential that the split reporter proteins do not spontaneously associate in the absence of binding partners (that normally drive the complementation). If so, a high number of false positives would be generated [112,128,129]. Ideally, when the site of interaction is known, a mutation at the specific interaction site of the receptor, is performed to evaluate whether this indeed disrupts the interaction. In those cases where the interaction site is unknown, a screening could be performed to elucidate the perfect non-interacting GPCR partner. Negative controls implemented in already published research articles are shown in Table 5 for the BiFC and in Table 6 for BiLC.

### 6.4. Normalization Factor

To allow better comparison between different experimental setups, it is advisable to include a normalization factor, i.e., a reporter gene such as a fluorescent marker, as previously described [112,147]. In a transient expression system, co-transfection of this fluorescent marker, which should not cause cross-excitation, is recommended. A list of PCAs with corresponding fluorescent markers is given in Table 7. For BiLC in principle, all fluorescent reporters can be used, although it is recommended to avoid cross-excitation by choosing an excitation wavelength more than 20 nm before or after the emission wavelength of the luminescent protein. A potential strategy to cope with alterations in expression levels of the fusion proteins, is the implementation of ratiometric expression systems, consisting of two GPCRs of interest and a fluorescent marker for optical expression control. The so-called ratiometric BiFC (rBiFC) makes it feasible to control the expression through FACS or flow cytometry analysis [147].

### 6.5. Endogenous Expression Levels

In an arbitrary system, PCAs often involve overexpression of split biosensors since these reporter genes are typically introduced in the cells by transfection. To circumvent excessively high levels of membrane expression, which may ultimately result in random collisions rather than a real interaction, low levels of biosensors should ideally be expressed, preferably close to endogenous expression levels. White et al. [148] (2017) have demonstrated that PPIs can be endogenously monitored by BRET when a Nanoluciferase reporter is genetically fused to a natively expressed GPCR of interest by CRISPR/Cas9-mediated homology-directed repair. This methodology also offers the potential to study GPCR–GPCR interactions at endogenous levels.

### 6.6. Kinetics

To follow the dynamics of GPCR interactions in living cells in real time or to monitor the influence of ligand-dependent modulation on the level of dimerization, one can use a real-time PCA-based method for a certain amount of time. For example, to evaluate the effect of the α-melanocyte-stimulating hormone (α-MSH) on the interaction between MC4R and MRAP2, Habara et al. [113] (2018) measured this interaction with a NanoBiT^®^-based PCA for up to 120 s before and after the addition of α-MSH. By doing so, the authors could demonstrate that the stimulation of MC4R with α-MSH slightly decreased the NanoBiT^®^ signal, which led to the postulation that the activated structural change of MC4R negatively impacts the interaction with MRAP2. Overall, depending on the half-life of the substrate and the PPI of interest, different timeframes of real-time measurement can be implemented.

## 7. HTS with Cell-Based PCAs

High-throughput screening (HTS) assays are powerful ways to assess the influence of ligands that affect the protein complexes, thereby providing a method to measure spatial and temporal changes in the protein association in response to drugs (Figure 7). This section will focus on the potential use of HTS assays based on BiLC and BiFC for GPCRs.

### 7.1. GPCR Oligomerization Screening

Given that the protein interaction partners are known, PCAs based on BiLC and BiFC could aid in studying the strength of the interaction compared to a negative control. To find new (unknown) interactions, a screening assay would be handy, in which case the bait protein is fused to a split luminescent/fluorescent fragment and screened against a cDNA library fused to the other half of the reporter fragment. This way, one can identify which GPCRs interact with each other. In conjunction to this is the functional validation of the detected interactions. False readouts can be sidelined upon employment of fusion proteins with mutations at the site of interaction, which consequently inhibit the interaction and decrease the level of complementation of the luminescent or fluorescent fragments. Clearly, this is only feasible if the interaction sites are known, which may be unraveled by alanine screening. For such screening assays, a robust plate reader format would be the best choice.

A non-GPCR example of such a workflow was published by Zych et al. in 2013 [149]. These authors established a high-throughput imaging based-screening that was based on the principles of BiFC, with as the aim the study of the dynamics of Vpr (a nonstructural protein encoded by HIV-1) dimerization.

Screening based on BiFC coupled to flow cytometry is a fast way to capture weak interactions, and hence, this has been used to screen for mutations that modulate the affinity and specificity of PPIs. For example, in 2013, Morell et al. [150] coupled flow cytometry to BiFC to study the weak interaction between the SH3 domain of C-Abl kinase and its natural or mutated binding partners. This combination helps to select for good interacting partners, even if they were deficient in the whole cell population. Hence, this combination has proven to be fast, specific, and sensitive and can capture even weak, transient interactions, thus opening up a new dimension in the field of proteomics.

### 7.2. GPCR Drug Discovery

Since GPCR–GPCR interactions have been implied in several disease patterns, e.g., neurodegenerative disorders, a strong interest in compounds which could interfere with or alter these GPCR–GPCR interactions has arisen. Monitoring these GPCR–GPCR interactions has the potential of unveiling differences in interaction specifics, such as dynamics, as well as identifying potential therapeutic agents.

For an in vitro HTS assay, PCAs can be used to screen for compounds that inhibit (BiLC) or enhance (BiLC and BiFC) the PPIs. Upon addition of the modulating compound, the recorded signal changes, depending on its potential to influence the GPCR–GPCR interaction. Counter screens should be implemented to negate the possibility of an artefact, such as compounds which possibly interfere with reporter complementation, show autofluorescence, scatter light, quench luminescence or have toxic side-effects [151,152,153]. A key advantage of PCA-based approaches is that these include formats using living cells, so GPCR receptors are in their native environment.

Overall, this type of HTS strategy would be of great help to further enhance the interest in GPCR drug-discovery research.

## 8. In Vivo Application

Besides the potential as HTS assays, in recent years, BiFC and BiLC complementation techniques have also been applied in vivo, aiming at a better understanding of the dynamics of homo- or heterodimers in living animals. Several studies established an imaging reporter strategy, which can monitor the specific pharmacological regulation of oligomer complexes.

For GPCR dimerization purposes, Luker et al. [98] (2009) applied the FLuc-based PCA (NFLuc-416 and CFLuc-398) to investigate the homo- and heterodimerization of CXCR4 and CXCR7 in a tumor xenograft model of breast cancer. Type 231 cells stably expressing CXCR4-NFLuc and CXCR4-CFLuc or CXCR7-NFLuc and CXCR7-CFLuc were injected bilaterally into 6 to 8 weeks-old female nude mice. To test whether treatment with chemokine ligands produced time- and dose-dependent changes in reporter signal due to the alterations in the level of dimerization, fibroblasts stably expressing the chemokine CXCL12 were co-implanted. This chemokine reduced the level of both CXCR4 homodimerization and CXCR4 − CXCR7 heterodimerization. Blocking the binding of CXCL12 to CXCR4 resulted in a time-dependent increase in the level of CXCR4 homodimer formation, but no increase in the CXCR4 − CXCR7 heterodimerization was detected. In another setup, mice were injected with 231 cells stably expressing CXCR4 or CXCR7 homodimers and with fibroblasts only transduced with GFP. After 1 h of treatment with the CXCR7 modulator CCX754 (100 mg/kg subcutaneous), an increase in CXCR7 homodimer formation, but not CXCR4 homodimer formation, was detected. Overall, it was observed that the chemokine CXCL12 decreased the level of CXCR4 homodimerization and CXCR4 − CXCR7 heterodimerization and the CXCR7 modulator CCX754 increased the level of CXCR7 homodimerization. The application of such newly designed drugs that specifically modify the levels of certain dimers could be beneficial in the treatment of not only breast cancer, but also in rheumatoid arthritis, HIV disease, lung and prostate cancers, where these GPCRs play an important role as well.

Rather than trying to bridge the gap between cell culture studies and in vivo physiology by the injection of cells stably expressing GPCRs fused to the split biosensors, use of the CRISPR/Cas9 methodology (previously described in Section 6.5) to fuse native receptors to split protein fragments would allow the detection of dimerization at endogenous protein levels. This has already been successfully implemented for the β-arrestin2 recruitment to CXCR4 [148], thus, it could likewise be tested for GPCR dimerization.

Most importantly, it has been shown that the bioluminescent reporter is sensitive at all the depths and locations within a live mouse [154], thus making this one of the best platforms to study PPIs as it better mimics a “real-life scenario”.

In the field of fluorescence, Han et al. [155] in 2014 applied BiFC based on split fragments of mNeptune, a far-red monomeric Neptune variant (600 nm/650 nm), for in vivo applications. The main obstacle of imaging in live animals, being the tissue opacity to excitation light below 600 nm, was overcome by this fluorescent protein. This system was tested by imaging the interaction between bFos and bJun in live mice and could also serve to unravel GPCR–GPCR interactions.

Finally, luminescence- or fluorescence-based systems can be applied to study GPCR oligomerization in vivo as these assays allow real-time detection of the interaction and subsequently would help in the screening of pharmacological compounds acting on these oligomers. These non-invasive animal imaging assays, for the quantification of GPCR–GPCR interactions, might aid in the transformation of treatments targeting specific oligomers (e.g., small molecules or bivalent ligands) from cell-based assays to clinical trials.

## 9. Conclusions

GPCRs are highly dynamic proteins that are subject to multiple spatiotemporal interactions inside the cell to trigger appropriate cellular responses. These interactions involve transducers, kinases, as well as other GPCRs, among others [156,157,158]. While great progress has been made during the past decade in targeting and understanding GPCR–GPCR interactions, some key questions remain unanswered. Although increasing evidence supports the existence of GPCR–GPCR interactions in vitro and in vivo [159], the sophisticated characteristics of these interactions, such as their variation in dynamics, complexity (i.e., dimerization or higher-order oligomerization), binding affinities, and the lack of a consensus sequence in the interfaces, make GPCR–GPCR interactions a challenging research field. Multiple studies based on PCA techniques have shed new light on the composition, localization, and even ligand-induced modulation of these GPCR complexes. Moreover, the potential of these PCA-based methodologies has been further explored by their combination in a variety of ways to investigate not only binary but also ternary or even quaternary partnerships. The current review describes the journey taken by these BiLC- and BiFC-based PCAs and the GPCR–GPCR interactions that have been identified by applying these techniques. PCAs are straightforward and flexible, and provide on the one hand a convenient approach to visualize the oligomerization processes in living cells and on the other hand an ideal tool to monitor the interaction dynamics. Nevertheless, certain milestones could still be achieved, such as the development of reversible split-fluorescent proteins or the discovery of split-luminescent proteins and their respective ligands that allow enhanced imaging of the GPCR–GPCR interactions. Finally, we anticipate a growing interest in the application of PCA-based high-throughput drug screenings, which makes it possible to identify compounds that selectively modulate the composition of GPCR oligomers in living cells.

## Figures and Tables

**Figure 1 ijms-20-02958-f001:**
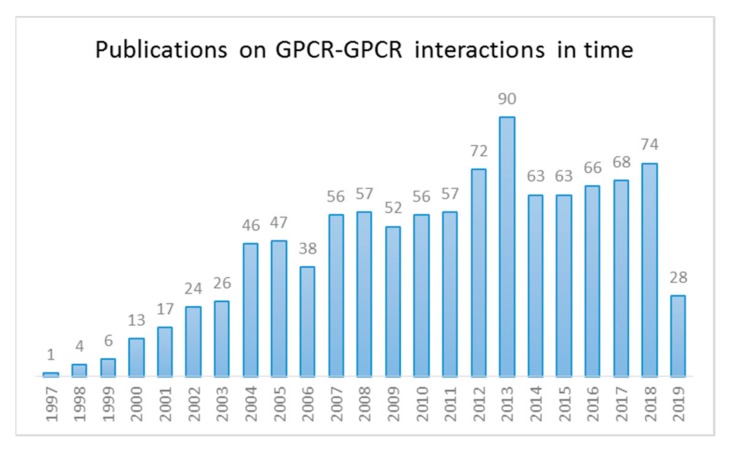
Number of publications on G protein-coupled receptor (GPCR) dimer and oligomers from 1997–2019 (data incomplete for 2019). (Source: NCBI—a search for the terms ‘‘GPCR* AND DIMER*’ or ‘GPCR* AND OLIGOMER*’’ in the PubMed database).

**Figure 2 ijms-20-02958-f002:**
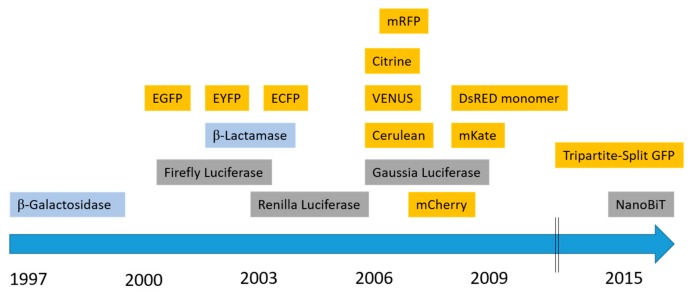
History of the development of split protein biosensors. Commonly used fluorescent split proteins (yellow), luminescent split reporters (grey), and enzymes with colorigenic substrates (blue) are shown.

**Figure 3 ijms-20-02958-f003:**
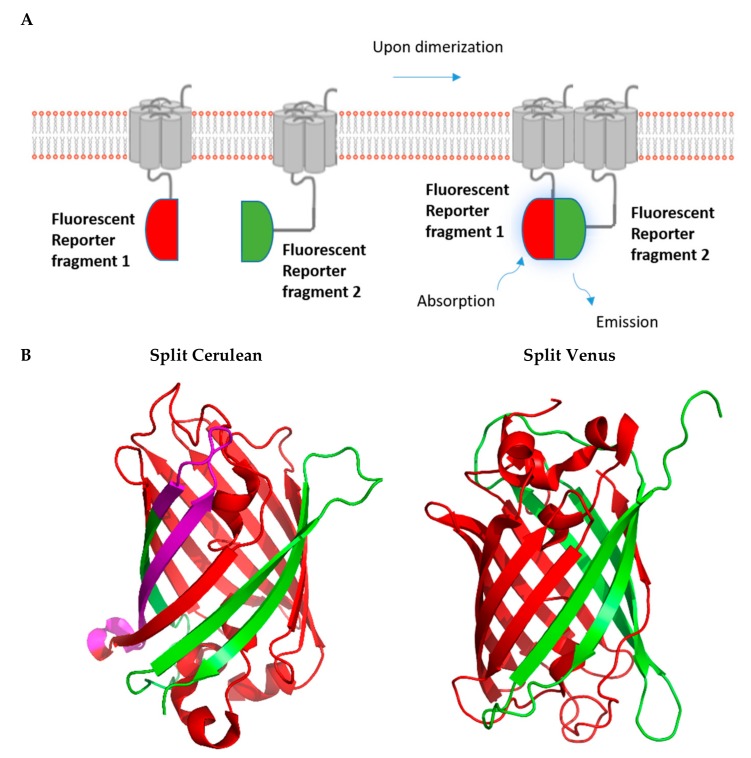
Fluorescence-based complementation assay. (**A**). The principle of the fluorescent complementation assay is shown schematically. (**B**). Split versions of the fluorescent reporters Cerulean and Venus are shown in green and red. Purple refers to overlapping sections. (PDB; Accession no. 3AKO for Venus; PDB: Accession no. 5OXB for Cerulean).

**Figure 4 ijms-20-02958-f004:**
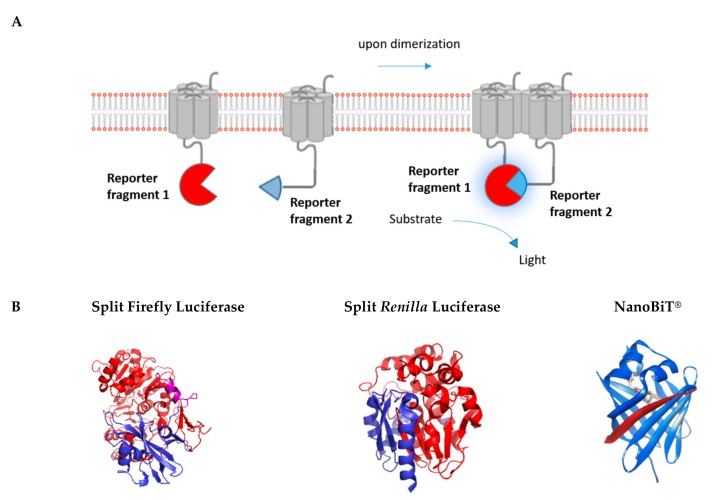
Luminescence-based protein complementation assay. (**A**) The principle of the luminescent complementation assay is shown schematically. (**B**) Split versions of the luminescent reporters Firefly, *Renilla*, and Nanoluciferase are shown in blue and red. Purple refers to overlapping sections. Split luminescent biosensors are depicted in proportion to their size (Fluc: 62 kDa, Rluc: 36 kDa, and NanoBiT^®^: 19 kDa) (PDB Accession no. 1LCI for Firefly luciferase, PDB Accession no. 2PSD for *Renilla* luciferase) (NanoBiT^®^: source Promega).

**Figure 5 ijms-20-02958-f005:**
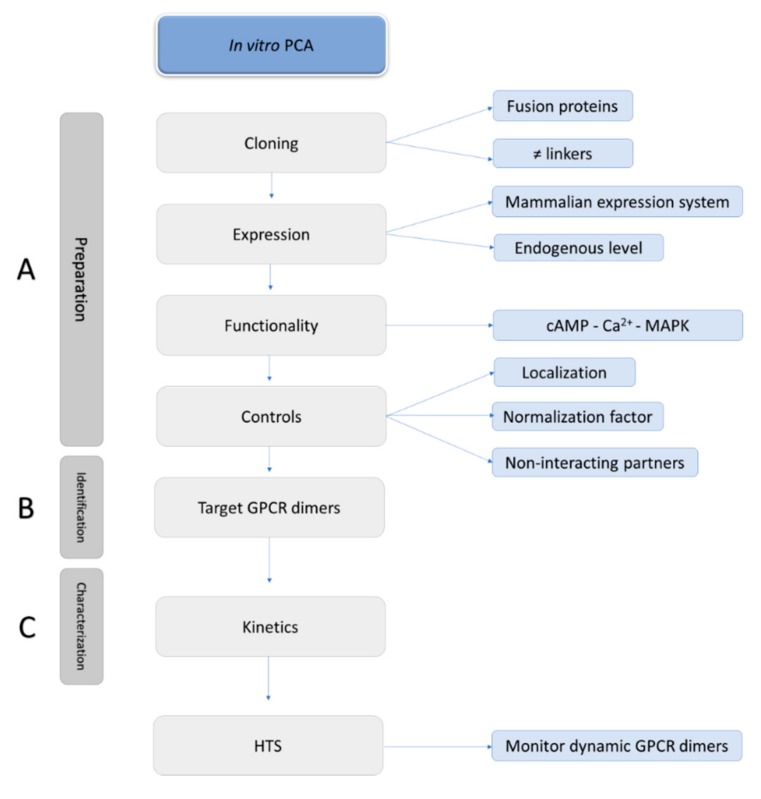
A guideline for the set-up of an optimized PCA.

**Figure 6 ijms-20-02958-f006:**
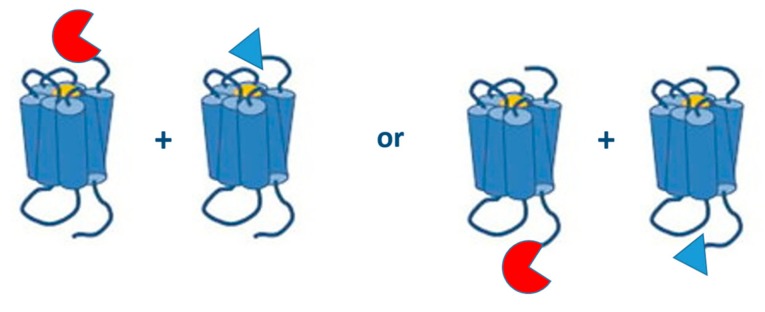
N- or C-terminally split biosensor GPCRs.

**Figure 7 ijms-20-02958-f007:**
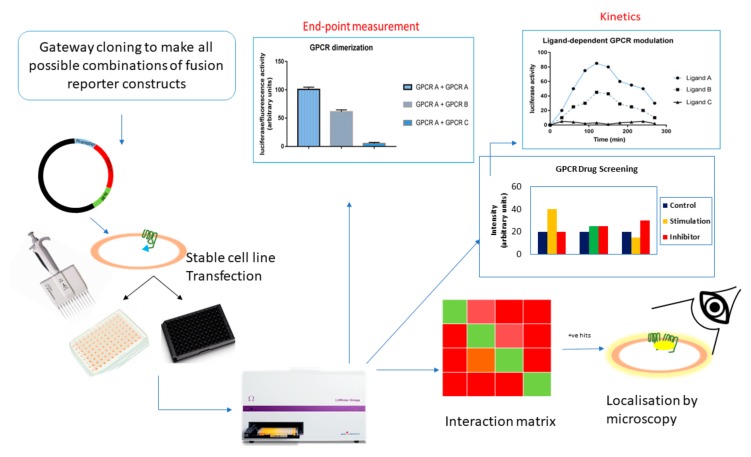
High-throughput screening.

**Table 1 ijms-20-02958-t001:** Overview of proteins, already implemented in fluorescence- and luminescence-based complementation assays. N/A = not applicable, nd = not determined.

Reporter Protein	Source Species	Readout	Excitation Wavelength (nm)	Emission Wavelength (nm)	Substrate	Cofactor	Stability (h)	Maturation Time (t1/2) (min)	MW (kDa)
VENUS	*Aequorea victoria*	Fluorescence	515	528	-	N/A	-	40 (in vitro)	27
GFP	*Aequorea victoria*	Fluorescence	488	510	-	N/A	-	53 (in vitro)	27
mCherry	*Discosoma*	Fluorescence	587	610	-	N/A	-	17 + 30 (*S. cerevisiae*)	29
Cerulean	*Aequorea victoria*	Fluorescence	433	475	-	N/A	-	nd	27
Tripartite-Split GFP	*Aequorea victoria*	Fluorescence	488	530	-	N/A	-	nd	23
EYFP	*Aequorea victoria*	Fluorescence	514	527	-	N/A	-	23 (in vitro)	26.4
ECFP	*Aequorea macrodactyla*	Fluorescence	405	485	-	N/A	-	49 (*S. cerevisiae*)	26.8
Citrine	*Aequorea victoria*	Fluorescence	516	529	-	N/A	-	nd	27
mRFP	*Discosoma striata*	Fluorescence	584	607	-	N/A	-	<60	25.9
mKate	*Discosoma striata*	Fluorescence	588	635	-	N/A	-	20	26
DsRed monomer	*Discosoma striata*	Fluorescence	558	583	-	N/A	-	600	28
Renilla luciferase	*Renilla reniformas*	Luminescence	-	480	Coelenterazine	N/A	4.5 h (cell)	-	36
Firefly Luciferase	*Photinus pyralis*	Luminescence	-	550–570	d-luciferin	ATP, O_2_	4.0 h (cell)	-	62
Gaussia Luciferase	*Gaussia princeps*	Luminescence	-	485	Coelenterazine	N/A	60 h (cell media)	-	20
NanoBiT	*Oplophorus gracilirostris*	Luminescence	-	460	Furimazine	N/A	6.0 h (cell)	-	19
β-lactamase	*Bacillus licheniformis*	Luminescence	-	492	Nitrocefin	N/A	nd	-	29
β-Galactosidase	*Escherichia coli*	Fluorescence	Reliant on the substrate	Reliant on the substrate	FDG, MUG a.o.	Mg^2+^	1.1 h (yeast cells)	-	464
Click Beetle luciferase	*Pyrophorus plagiophthalamus*	Luminescence	-	-	d-luciferin	Mg^2+^, ATP	-	-	64

**Table 2 ijms-20-02958-t002:** Combinatorial assays with BiFC and BiLC for GPCR oligomerization purposes.

GPCR Dimer	Oligomeric Type	PCA Type	Split Biosensor	Fragments	Negative Control	Cell-Type	Year	Ref.
CXCR4 − CXCR4/CC2	Hetero-oligomer with CC2, homotetramer	BiLC and BiFC	*R*LucII, vYFP	*R*LucII: 1–330, 331–936 vYFP: 1–465, 466–720	D_2_R	HEK293	2014	[99]
A_2a_ − D_2_R	heterotetramer	BiLC and BiFC	*R*Luc8, YFP	*R*Luc8: 1–229, 230–311 YFP: 1–155, 156–238	A_1_R, D_1_R	HEK293	2015/2016	[101,118]
D_2S_R − D_2S_R	Homo-oligomer	BiLC and BiFC	*R*Luc8, mVenus	*R*Luc8: 1–229, 230–311 mVenus: 1–155, 156–240	CD8, TSHr	HEK293T	2008	[100]
β_2_AR − β_2_AR	homotetramer	BiLC and BiFC	*G*Luc, Venus	*G*Luc: 1–63, 64–185	*G*LucN, VN, VC	HEK293	2008	[77]

**Table 3 ijms-20-02958-t003:** Advantages and disadvantages of BiFC and BiLC techniques.

BiFC	
Advantages	Disadvantages
Straightforward technique	Need for tagged proteins (GPCRs)
High-throughput experiments	Autofluorescence
Imaging microscopy: localization of the interaction	Photobleaching
Study intact cells	Measuring dynamics: limited (maturation time) Not applicable for studying inhibition of interactions
**BiLC**	
**Advantages**	**Disadvantages**
Straightforward technique	Need for tagged proteins (GPCRs)
High-throughput experiments	Requires a substrate
Kinetic measurements	Detection of localization: limited
Study intact cells	
In vivo application	

**Table 4 ijms-20-02958-t004:** Linker sequences applied in PCA assays for studying GPCR dimerization.

GPCR Dimer	PCA Type	Linker	Ref
**AT_1_ − AT_2_**	BiFC, Venus	GGGGSGGGG	[125]
**CXCR4 − CXCR4**	BiLC, *R*luc	(GGGS)2	[99]
**D_2L_R − D_2L_R**	BiFC, Venus	LG	[100]
**D_2L_R − D_2L_R** **A_2a_ − D_2_R**	BiLC, *R*luc	ATGLDLELKASNSAVDGTAGPVAT	[112,117]
**D_2L_R − D_2L_R**	BiLC, NanoBiT	GNS-GSSGGGGSGGGGSSG	[112]
**MOP − NPFF2**	BiFC, Venus	DGGSGGGS	[123]

**Table 5 ijms-20-02958-t005:** An overview of GPCR–GPCR interactions, shown by fluorescence complementation assays.

GPCR Dimer	Oligomeric Type	PCA	Split Biosensor	Fragments	Negative Control	Cell-Type	Year	Ref.	In Vivo or Native Tissue Evidence	Ref.
mGluR_5_− D_2_R	Heterodimer	BiFC	YFP	1–155, 155–231	GABA_B2_	HEK	2009	[130]	Yes	[130]
D_2_R − D_2_R	Homodimer	BiFC	YFP	1–155, 156–238	D_1_R	HEK	2015	[118]	Yes	[131,132]
A_2A_ − D_2L_R	Heterodimer	MBiFC	Venus/Cerulean	1–172, 155–238	D_1_	CAD	2008	[59]	Yes	[130,133,134,135,136,137]
D_2L_R − CB_1_	Heterodimer	MBiFC	Venus/Cerulean	1–172, 155–238	M_4_	CAD	2010	[62]	Yes	[133]
D_2L_R − D_2L_R	Oligomer	MBiFC	Venus/Cerulean	1–172, 155–238	-	CAD	2010	[62]	-	-
D_2S_R − D_2S_R	Homodimer	BiFC	Venus	1–155, 156–240	CD8	HEK293T	2008	[100]	Yes	[131,132]
AT_1_ − AT_2_	Homo- and heterodimer	BiFC	Venus	1–158,159–239	ATIP	HEK293FT	2011	[125]	Yes	[138,139,140]
CXCR4 − CXCR4	Homodimer	BiFC	vYFP	1–465, 466–720	D_2_R	HEK293	2014	[99]	-	-
A_2A_ − A_2A_	Homodimer	MBiFC	Venus/Cerulean	1–172, 155–238	-	CAD	2008	[59]	-	-
A_2A_ − A_2A_	Homodimer	BiFC	YFP	1–155, 155–238	Non-fused A_1_ (competition)	HEK293T	2018	[65]	-	-
A_2A_ − A_1_	Heterodimer	BiFC	YFP	1–155, 155–238	-	HEK293T	2018	[65]	Yes	[141]
GHSR1a-OX1R	Heterodimer	BiFC	Venus	1–172, 156–239	-	HEK293T	2018	[64]	-	-
β_2_AR − β_2_AR	Oligomer	BiFC	− 15sfGFP		-	HeLa	2019	[67]	-	-
A_2A_ − A_2A_	Homodimer	BiFC	YFP	1–155,156–239	GABA_B2_	CHO	2018	[66]	-	-
A_2B_ − A_2A_	Heterodimer	BiFC	YFP	1–155,156–239	GABA_B2_	CHO	2018	[66]	-	-
FFAR3 − FFAR3	Homodimer	BiFC	Venus	1–155 (I152L), 155–239	P2RY1	HEK293T	2018	[63]	-	-
FFAR2 − FFAR3	Heterodimer	BiFC	Venus	1–155 (I152L), 155–238	P2RY1	HEK293T	2018	[63]	-	-
mGluR_2_ − mGluR_2_	Homodimer	BiFC	mCitrine	1–172, 155–238	-	HEK293T	2016	[142]	-	-
α_1b_ − α_1b_	Homodimer	BiFC	eYFP	1–172, 155–238	-	HEK293T	2007	[143]	-	-

**Table 6 ijms-20-02958-t006:** An overview of GPCR–GPCR interactions, shown by luminescence complementation assays.

GPCR Dimer	Oligomeric Type	PCA Type	Split Biosensor	Fragments	Negative Control	Cell-Type	Year	Ref.	In Vivo or Native Tissue Evidence	Ref.
CXCR4 − CXCR4/CC2	Homodimer	BiLC	FLuc	NLuc-416 and CLuc-398	β_2_-AR	HEK293T	2009	[98]	Yes	[144,145,146]
CXCR7 − CXCR7	Homodimer									
CXCR4 − CXCR7	Heterodimer									
A_2a_ − D_2_R	heterodimer	BiLC	*R*Luc*8*	1–229, 230–311	-	HEK293T	2018	[117]	Yes	[118,130,134,135,136,137]
A_2a_ − A_2a_	homodimer	BiLC	*R*Luc*8*	1–229, 230–311	A_1_	HEK293	2016	[101,118]	-	-
D_2L_R − D_2L_R	homodimer	NanoBiT	NanoLuc	1–11, 12–167	CB_2_	HEK293T	2018/2019	[40,112]	Yes	[131,132]

**Table 7 ijms-20-02958-t007:** The application of normalization factors. * Given as examples.

Fluorescence	PCA (Excitation/Emission)	Fluorescent Marker (Excitation/Emission)
	Venus _(515/528)_	CFP _(433/475)_
	mCherry _(587/610)_	CFP _(433/475)/_GFP _(488/510)/_YFP _(514/527)_
	GFP _(488/510)_	CFP _(433/475)_
	Cerulean _(452/478)_	mTagBFP _(402/457)_
Luminescence	PCA	Fluorescence Marker
	*R*luc	CFP _(433/475)/_mCherry _(587/610)_ *
	Fluc	CFP _(433/475)/_mCherry _(587/610)_ *
	Nluc	CFP _(433/475)/_mCherry _(587/610)_ *

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
