# Peer review of "Luminescence- and Fluorescence-Based Complementation Assays to Screen for GPCR Oligomerization: Current State of the Art"

_ijms, 2019, doi:10.3390/ijms20122958_

Round 1
Reviewer 1 Report
This review provides an overview of fluorescent and luminescent protein complementation assays oriented for the studies of GPCR dimerization/oligomerization. This manuscript comprehensively contains the history, principle, design, applications, and implementations. Especially, the comparison of methodologies of available PCAs and list of advantages and drawbacks are useful for the researchers who are using or planning to use PCAs. This will also encourage the many researchers in molecular science field to consider PCAs for their studies.
While this review is totally excellent, my only concern is the introduction of rhodopsins (line 94). The dimeric structure of rhodopsin in native membrane was clearly demonstrated, and many researchers still claim that the oligomerization of rhodopsin is essential for its function. I prefer more balanced description for the history and physiological relevance of rhodopsin oligomerization.
Author Response
We wish to thank the reviewer for the detailed reading, the constructive comments and for the overall positive assessment of our manuscript. By complying to all reviewer’s comments we feel that the quality of our manuscript has further improved.
The response to the reviewer is uploaded as a PDF file.

Reviewer 2 Report
Nice review article about BiFC assays
Minor comments:
Another two examples of BiFC studies with GPCR complexes that should be included in Table 4:
alpha1b oligomer:
https://www.ncbi.nlm.nih.gov/pubmed/17220353
5-HT2A-Glu2 heterocomplex
https://www.ncbi.nlm.nih.gov/pubmed/26758213
Dopamine D2 oligomer
https://www.ncbi.nlm.nih.gov/pubmed/18668123
Author Response

(The authors gave the same response as above.)

Reviewer 3 Report
This is an excellent review of the current state of the field of bioluminescence/fluorescence-based complementation assays for the study of GPCR oligomerization. Just a couple of comments to help the authors further improve the quality of their manuscript:
1) It is important to include a short section (1-2 paragraphs long) listing all the GPCR oligomerization examples discussed in the manuscript that have been verified to naturally occur in vivo, e.g. via co-IP [as described in: Methods Cell Biol. 2019;149:205-213 & in: Biosci Rep. 2017;37(2)]. This will help the reader to distinguish the really important (physiologically relevant) examples of the GPCR oligomerization phenomenon from others that have been reported only in recombinant systems in vitro and thus, might be technical artifacts lacking real physiological relevance.
2) Although not exactly a form of GPCR-GPCR dimerization, a brief discussion of the properties of RAMPs (receptor activity-modifying proteins) and of how they interact with certain GPCRs to modulate their functions, including the major applied examples of this: adrenomedullin, calcitonin family peptide, and CGRP receptors (e.g. see: Peptides. 2019;111:55-61), should be included in "Introduction".
3) The English of the manuscript needs some polishing. For example, line 87: "These receptors are obliged to form a dimer..." might be better off rephrasing to: "These receptors are obligatory dimers...", etc.
Author Response

(The authors gave the same response as above.)
